# Preparation of Alginate-Based Biomaterials and Their Applications in Biomedicine

**DOI:** 10.3390/md19050264

**Published:** 2021-05-10

**Authors:** Hengtong Zhang, Junqiu Cheng, Qiang Ao

**Affiliations:** 1National Engineering Research Center for Biomaterials, Sichuan University, Chengdu 610064, China; zhanght_1226@163.com; 2NMPA Key Laboratory for Quality Research and Control of Tissue Regenerative Biomaterial, Institute of Regulatory Science for Medical Devices, NMPA Research Base of Regulatory Science for Medical Devices, Sichuan University, Chengdu 610064, China

**Keywords:** alginate, hydrogel, wound dressing, drug delivery, tissue engineering

## Abstract

Alginates are naturally occurring polysaccharides extracted from brown marine algae and bacteria. Being biocompatible, biodegradable, non-toxic and easy to gel, alginates can be processed into various forms, such as hydrogels, microspheres, fibers and sponges, and have been widely applied in biomedical field. The present review provides an overview of the properties and processing methods of alginates, as well as their applications in wound healing, tissue repair and drug delivery in recent years.

## 1. Introduction

Alginates are groups of linear anionic polysaccharides derived from kelp or *Sargassum* algae of brown algae and several bacterial strains [1]. Since its discovery in the late nineteenth century, alginates have been researched extensively on their physical and chemical properties [2,3,4]. Alginates are non-toxic, rich in source and easy to obtain. It has been proved to be biocompatible and biodegradable in human body [5,6,7]. Based on these favorable properties, alginates have been widely used in the food industry, and have become very important biomaterials for phamaceutical and biomedical purpose.

This review covers the recent advances on the characterizations and productions of alginates, and presents several important fabrications of alginate biomaterials, including hydrogels, microspheres, and composite porous scaffolds. Besides, the review describes the versatile applications of the alginate-based materials in protein and small molecule drug delivery, wound dressing, skin, cartilage, bone tissue repair and regeneration, and 3D bioprinting as well.

## 2. Compositions, Structures and Properties

Chemically, alginates are copolymers mainly composed of β-d-mannuronic acid (M) and its C5 epimer α-l-guluronic acid (G) residues linked via 1,4-glycosidic bond in an irregular block-wise manner. The M and G residues are organized in homopolymeric blocks of G units (GG blocks) or M units (MM blocks) and heteropolymeric sequences of randomly coupled G and M units (GM or MG blocks). Their occurrence, proportions and distributions may differ significantly depending on their natural sources [2,3,4].

Due to the steric hindrance around the carboxyl groups in the ring structure of the residues, M units adopt a stable 4C_1_ chair conformation, while ^1^C_4_ chair conformation is preferred for G units, so that the spatial interference of the groups present in the ring structure could be reduced [8]. The tacticity of chemical groups around M and G units results in the formation of different molecular conformations. The linkages in diequatorial position of the MM blocks, in diaxial position of the GG blocks, and in equatorial/axial or axial/equatorial position of the MG or GM blocks in the copolymer (Figure 1), give rise to a flat ribbonlike structure for the MM blocks, a rigid folded structure for the GG blocks and a helix-like structure for the MG or GM blocks, respectively [9,10].

The compositions and conformation structures of the blocks as well as their arrangement patterns determine the physicochemical properties of the alginates, which in turn have both biological and industrial significance.

### 2.1. Molecule Rigidity/Flexibility, Solubility and Viscosity

The linkage in the block structure results in varying degrees of stiffness or flexibility in alginates, due to the greater or lesser hindrance of the rigid six-membered sugar rings and restricted rotation around the glycosidic bonds, as well as the electrostatic repulsion between the charged groups on the polymer chain [9,10]. As the relative flexibility increases in the order GG block < MM block < MG or GM block, the polymer chains of alginates containing predominantly GG blocks are stiffer and possess a more extended chain conformation than those containing MM blocks, which in turn are stiffer than MG or GM blocks, and the rigidity increases in the order MG < MM < GG [11].

Alginic acid is insoluble in water and organic solvents, whereas its monovalent salts and esters are water-soluble and form a stable, viscous solution [12]. The intrinsic viscosity of alginates is determined by molecular weight, rigidity, extension of the chain of the polymer and ionic strength of the solution. The increase in molecular weight, alginate concentration, or the stiffness of the constituent chain blocks leads to the increase in the viscosity. Besides, with the extension of polymer chain, the electrostatic repulsion between the charged groups on the chain increases and the intrinsic viscosity follows with an increase [11].

### 2.2. Gelation Properties

The most important property of alginates is their ability to form ionic gel in the presence of polyvalent cations. The gelling is the result of ion-exchange between monovalent ion of alginate solution (most often, sodium ion, Na^+^) and polyvalent cations followed by the subsequent coordination of polyvalent metal ions with alginate macromolecule. Calcium is the most common cation applied to form ionic alginate gels.

The typical gelation mechanism involves the coordination and chelating structures in the model of egg-box during the process of binding of alginates to polyvalent metal ions, G-units selectively form higher-order junction zones, which is composed of two or more chains, together with the hydrogen-bonding interaction of these cross-linking agents with oxygen atoms in the G blocks of two adjacent polymer chains (Figure 2) [13,14]. In the 3-D network of the egg box, each cation is bound with four G residues, thus there should be eight to 20 adjacent G residues in order to form a stable junction for Ca alginate gels [13].

Continuous revision and improvement on the gelling mechanism has been carried out based on new experimental evidence. Further work has suggested that the egg box model is not the only possible structure for the junction zones, a 3/1 helical conformation of G blocks is more proper for Ca alginate gels formed slowly, while the 2/1 helical conformation is still possible in fast gelatinized Ca-alginate in which the 2/1 helix is a metastable form [15]. It has been accepted that Ca-alginate gelation undergoes three distinct and successive steps, which starts with the interaction of Ca^2+^ with a single G unit to form monocomplexes in a tilted egg-box structure, followed by the pairing of the monocomplexes to form egg-box dimers, and then lateral association of the egg-box dimers to generate multicomplexes, which is mediated by Ca^2+^ concentrations, disordered Na^+^, water molecules and hydrogen bonding between the hydroxyl and carboxyl groups of paired G residues [13,16,17,18]. The alginate chains are auto cooperatively zipped by inter-cluster separation and simultaneously intra-cluster association of egg-box dimers, which is potentially attributed to the excessive Ca^2+^ neutralizing the free negative carboxyl groups in alginate molecules which disrupts the association between egg-box dimers [19].

The alginate gel formation is determined by the type and concentration of cross- linking agents/cations, as well as the molecular weight, composition, degree of polymerization and polymer chain length of the alginates. The strength and viscosity of the formed gels are the two most important physical properties that denote the gelling capability of alginates.

Previous studies have shown that several divalent cations can bind to alginates, but with different affinity, which is in the order of Mg^2+^ < Mn^2+^ < Zn^2+^, Ni^2+^, Co^2+^ < Fe^2+^ < Ca^2+^ < Sr^2+^ < Ba^2+^ < Cd^2+^ < Cu^2+^ < Pb^2+^ [15,20]. Harper et al. explored the effect of the gelling cations on the properties of the resulting alginate gels on the basis of ionic radii. With the addition of Ba^2+^, Sr^2+^, Ca^2+^, Zn^2+^, and Mg^2+^ respectively in the sodium alginate solution, the corresponding gel strength expressed in the Young’s modulus showed that Ba^2+^ formed the strongest gels followed by Sr^2+^, Ca^2+^, Zn^2+^, and Mg^2+^, which partly supported the presumption that cation size may influence the strength of gels in that larger cation might provide stronger binding forces at junction zones thereby creating stronger gels [21]. Similar conclusion can also be drawn from another research on the coordination model of Ca and Sr with alginates. Zhang et al. prepared Ca and Sr-alginate fibers with microfluidic spinning technology, and verified that the chelation type of Sr^2+^ with alginate being similar to that of Ca^2+^, i.e., core-shell of the analogous egg-box structure. Compared to Ca-alginate counterpart, there were more crosslinking sites of Sr^2+^ with alginate molecule, which resulted in higher crosslinking degree and stronger binding of Sr^2+^ with alginate molecular chain, hence more robust mechanical performance of Sr-alginate fibers [22]. Cation charges may also affect the gel properties. Trivalent cations such as Al^3+^ and Fe^3+^ can also be used to gel alginates. Due to their ability to bind with three carboxyl groups from different alginate polymer chains at the same time, they generally have an increased affinity of binding with alginates and form a more compact gel network by binding in a 3-D structure [20]. Generally speaking, many other factors influence the affinity of a certain metal cation with a polymer network besides its interaction with one functional group, including ionization potential, ionic charge, mass and radius of the metal ion, participation of inner orbitals in metal-ligand binding, interaction with adjacent hydroxyl groups/coordination number, and covalent bonding [23].

As to the important role the alginate polymer plays in the gelling, the M/G ratio, block-structure and degree of polymerization of the alginate backbone profoundly impact the gelling behavior of alginates [11]. Increasing the content of G residues in the chains, and especially, increase the length of homoguluronic block structures (i.e., FGG and FGGG) with the average G-block length larger than one (NG > 1), have been considered to correlate positively to gelling properties of alginate [13]. As MM blocks and alternating MG blocks have lower affinity toward the cation, M-rich samples often produce more soft and elastic gels. Raising the G block content or molecular weight of alginates both contribute to achieve more strong and brittle gels [12]. Especially, it is important to note that G blocks in fact are not the only sequences involved in junction formation, but that long alternating sequences also play an important part in the alginate gel network by forming MG-MG and mixed MG-GG junctions [20].

Several studies have reported poor gelling properties of the *Sargassum* alginates which have M/G ratios around or below one, along with high contents of GG-blocks [11]. Rhein-Knudsen et al. proposed an explanation to the discrepancy. Due to the partial hydrolysis in the experiment, the *Sargassum* spp. samples may have shorter guluronic acid rich alginate chains that are more prone to de-polymerization than the other samples, and *Sargassum* spp. alginate chains are probably not long enough to achieve high gel strengths. Thus, it is no strange that the relationship between the degree of polymerization of the acid tolerant moieties in the alginates (DPn) and the gel strengths showed a positive linear correlation. Low DP alginate from *Sargassum* spp. demonstrated low gelling strength. Thus, the degree of polymerization of the acid tolerant alginate backbone fragments, but not M/G ratio or homoguluronate dimer and trimer element contents, appeared to correlate to the alginate gel strength [11].

### 2.3. Immune Response

The immune response of alginates has always been of huge concern, though there is still an ongoing debate on their biological reactions in vivo. M-blocks has once been identified as the major initiator of the foreign body reaction, it has also been reported that macrophages and neutrophils infiltrated around the alginate capsules within two weeks after implantation, and collagen formed around the capsules with higher densities adjacent to the alginate-body interface [24]. Whereas little or no immune response to other alginate implants were found, especially the alginates of high purity caused no obvious foreign body reaction in vivo [25]. The immunogenic response at the injection or implantation sites might be attributed to impurities remaining in the alginate, such as heavy metals, proteins, endotoxins, polyphenolic compounds, etc. [26]. Achieving a suitable level of biocompatibility requires highly purified alginate.

Several factors are likely to affect the in vivo biological behavior of the alginates, including physicochemical properties, chemical compositions of alginates, the administration method, etc. [27]. F Ge et al. examined three types of alginate materials, respectively, with low viscosity, high viscosity and in particulate, by both in vivo and in vitro analyses. Administration of alginates by intra lymph node injection (I.L.N.) yielded more potent cytokine productions than other injection routes. Alginate materials, especially in particulate form, have the potential to be applied in inflammation related diseases [27]. M Bochenek et al. tested, in non-human primate (NHP) models, seven alginate formulations that were efficacious in rodents, including three that led to transient islet graft function in clinical trials. All formulations elicited significant foreign-body response (FBR) and pericapsular fibrotic overgrowth (PFO) one month post implantation; however, three chemically modified, immune-modulating alginate formulations elicited a reduced FBR. In conjunction with a minimally invasive transplantation technique into the bursa omentalis of NHPs, the most promising chemically modified alginate derivative (Z1-Y15) protected viable and glucose-responsive allogeneic islets for 4 months without the need for immunosuppression. Chemically modified alginate formulations may enable the long-term transplantation of islets for the correction of insulin deficiency [28].

## 3. Preparation and Processing Technology of Alginates

### 3.1. Preparation

Commercially available alginates are currently extracted from raw algae. Generally, the production consists of several steps, through which the water-insoluble mixed salts of alginic acid present in the brown seaweed cell walls are extracted and converted to soluble sodium alginates, eventually obtained as purified alginic acid or salts. The first step is usually the pretreatment with 0.1% formaldehyde to avoid pigments in alginates, and then dissolve with dilute acid, usually hydrochloric acid (HCl), to remove the counter ions of alginates, get rid of acid-soluble impurities and increase the yield of alginate as well [29]. In the second step, alkaline solution is added to the harvested insoluble alginic acid, thereby obtaining sodium alginate in aqueous solution. Sodium carbonate (Na_2_CO_3_) treatment is normally adopted in view of the potential harm the sodium hydroxide (NaOH) could have to the environment [26,30]. In the third step, after filtration, the sodium alginate solution can be precipitated into sodium alginate, calcium alginate or alginic acid directly by respectively adding in ethanol, calcium chloride or dilute hydrochloric acid (HCl) [30]. The precipitates will go through further purification and conversion as needed and then be separated, dried and milled [31]. In industry, however, an alternative extraction process for sodium alginate is often adopted using calcium chloride to precipitate first due to the higher binding affinity of calcium ions, and then Ca^2+^ from the alginate is exchanged for sodium ion to form sodium alginate (Figure 3).

The consecutive addition of the chemicals during the extraction process can affect the conversion and influence the yield and physicochemical properties of the isolated alginates. There exists a need for milder and more efficient extraction processing. Enzymatic extraction techniques of algal alginates have been studied, but not standardized to routine extractions. Enzymes such as alginate lyase, laminarinase, are used to degrade the seaweed cell wall, thus free alginate can be released. An example of optimized extraction processes of alginates from laminaria by hydrolysis of cellulase, pectinase and protease is as follows [32]: the first, hydrolyze the alginate samples with 2% cellulase at 55 °C and pH 4.5 for 20 min, 1% pectinase at 60 °C and pH 4.5 for 1.5 h and 1% protease at 80 °C and pH 8.0 for 3 h. And then inactivate all the enzymes by boiling solution for 10 min. Finally, add 1 mol/L CaCl_2_ solution to the hydrolysis solution with the volume ratio of 1:4 (the hydrolysis solution: CaCl_2_ solution). In the end, harvest the alginates after bleaching, acidification, drying and alkalization.

The effect of different extraction methods on the molecular structure and the bioactivity of the obtained alginates has been evaluated. NJ Borazjani et al. compared the alginates from *Sargassum angustifolium* treated by water, acid, alcalase and cellulase. The results showed that the use of enzymes considerably reduced protein (from 14.58% to <0.4%) and polyphenol (from 16.0% to <1.7 mg GA/g sample) contaminations of alginates compared to those of water and acid, and the FT-IR spectrum revealed that the extraction method did not affect the structure of the recovered alginates. The highest molecular weight (Mw) (557.1 × 103 g/mol) was found in acid treated alginate while the Mw of cellulase assistant alginate (356.2 × 103 g/mol) was the minimum. The SVg values varied from 2.79–5.17 cm^3^/g revealing the loosed conformational structures of alcalase and cellulase assistant alginates. Alcalase assistant alginate stimulated RAW264.7 cells to release nitric oxide and inflammatory cytokines TNF-α, IL-1, IL-6, IL-10 and IL-12, while enzyme-treated alginates showed maximum DPPH radical scavenging activity and reducing power. Hence, the study revealed the determinant effect of pretreatment during the extraction process of alginate and the beneficial influence of enzymatic process when biological functions of alginates are of high interest in the industry [33].

Alginates can also be synthesized by bacteria, and mainly from two strains, i.e., *Pseudomonas aeruginosa*, the pathogenic bacteria, and *Azotobacter vinelandii*, a nitrogen fixing soil-dwelling bacteria. Briefly, the biosynthesis process undergoes four stages: (1) precursor synthesis, (2) polymerization and cytoplasmic membrane transfer, (3) periplasmic transfer and modification, and (4) export through the outer membrane [34]. Although the two bacteria share a similar biosynthesis gene cluster, they differ in regard to epimerization as well as regulatory mechanisms, and produce alginates with different material properties. *Azotobacter* alginates contain all types of block structures, while *Pseudomonas* alginates only possess M and MG blocks, no G blocks, indicating their different biological role for different species [34]. Both bacterial alginates are characterized by acetylation of M-residues to a variable extent at positions O-2 and/or O-3. The degree of acetylation is also found to affect the material properties of the alginate. The presence of O-acetyl groups in bacterial alginates increases the interaction of chains with water molecules, leads to increasing water capacity and chain expansion and hence better solubility, notably changes the properties of the produced alginates in polymer conformation, viscoelasticity, and molecular mass [35].

Algal and bacteria alginates differ substantially from each other with respect to their composition, modification, molecular mass, viscoelastic properties, and polydispersity. Algal alginates are not naturally acetylated, usually contain all types of blocks including G blocks. The in vitro chemical modifications and treatments are normally required for alga alginates to obtain desired derivatives, which, however, are often less controllable and sometimes impossible, may result in undesired changes, such as degradation of polymer chain. Whereas the production by bacterial fermentation can be controlled and optimized on a large scale without geographical restriction and climatic influence. In addition, by adjusting the fermentation conditions such as temperature, pH and concentration of culture solution, it is possible to rationally design the synthesis for tailor-made alginates [36].

### 3.2. Hydrogels

Hydrogels are three-dimensional cross-linked networks of hydrophilic polymers [37,38,39], and many biomedical applications of alginates are in the form of hydrogels, including wound healing, drug delivery and tissue engineering [5,14,40,41]. Since the first report of its utility in insulin microencapsulation in 1980, alginate hydrogel has become the most widely used functional material for cell encapsulation and drug carrier [42]. As the ratio of M/G of the alginates from different sources varies, the G unit in alginate can provide rigidity to the polymer structure [43], and the pore size distribution of alginate hydrogel ranges widely, alginate hydrogels with different structures and functional characteristics can be prepared. For example, macroporous structure helps cells to effectively obtain nutrients, transmit metabolic waste and active substances, such as insulin and various growth factors. By choosing alginates with higher G unit content, the wider internal pore area of the alginate gel can be achieved.

Alginate hydrogels can be prepared through a number of methods, including ion interaction, covalent crosslinking, thermal gelation, and cell crosslinking [2,3]. By altering the type and density of crosslinking, the physical and chemical properties of the alginate hydrogels can be tailored for various biomedical applications [44]. Table 1 lists some of the typical alginate-based hydrogel matreials.

The most common method of alginate gel formation is ionic crosslinking with multivalent cations, which can be carried out under mild conditions [3]. Numerous researches have reported using Ca^2+^ as the ionic agent to prepare the alginate based hydrogel [45], with the major limitation that the formed gel will disintegrate in physiological environment. Strontium (Sr) is structurally, physically and chemically similar to calcium. It has been proved to have treatment effect on osteoporosis, thus become another commonly selected ion for alginate hydrogel [22]. As long as Sr only has stable isotopes (naturally occurring), the health risk is minimal.

Covalent crosslinking of alginate hydrogels forms the network through copolymerization or polycondensation reaction initiated by the crosslinking agent, which can improve the physical properties of hydrogels. The mechanical properties of the gels are mainly controlled by the cross-linking density and the agent type [46]. While the agent may be toxic to the cells or tissues in vivo, it needs to be removed completely after the hydrogel is formed. Composite hydrogels with both ionic and covalent cross-linking networks can also be made, and their mechanical properties can be improved through the synergistic effect of the two cross-linking mechanisms, which expands their applications [47].

The strategy of cell cross-linking alginate hydrogels is to form the network using specific receptor-ligand interactions. Alginate is composed of inert monomers and lacks the bioactive ligand required for cell anchoring and adhesion. The ligand (Arg-Gly-Asp, RGD) sequence is then introduced into alginate by chemical coupling via water-soluble carbon diamine chemical method [48,49]. When cells are dispersed in the RGD-modified alginate solution homogeneously, the receptors on the cell surface can be combined with the ligand in the modified alginate and thus form a cross-linked network [50]. Such cell-crosslinking hydrogels exhibit excellent bioactivity, and may be an ideal choice for cell delivery in tissue engineering, but currently the studies in this area are quite few.

In addition, thermal response phase transition has also been widely used in the alginate hydrogels preparation. Thermo-sensitive hydrogels used in drug delivery can adjust their swelling properties in response to temperature changes, thereby controlling the release of drugs from the gel as on demand. Poly(N-isopropylacrylamide) (PNIPAAm) hydrogels are the most extensively exploited thermo-sensitive gels, by incorporating it into the alginate hydrogel framework, the alginate hydrogel exhibits temperature-dependent behavior [51,52,53]. Spontaneous self-assembled physical interactions are also widely used in the formation of physical networks. Zhao et al. used the in-situ multilayer self-assembly technology to combine alginate with highly hydroxyl grouped polyacrylamide, and constructed a highly self-healing hydrogel with ordered semi-interpenetrating polymer network (semi-IPN) [54]. The hydrogel possesses a self-healing ability of 99% with being sprayed by only a small amount of water. Moreover, the layered semi-IPN structure leads to the tensile strength of PAMSA hydrogel up to 266 kPa.

Alginate hydrogel can be prepared by enzymatic cross-linking [55]. Enzymes are highly efficient and specific, can catalyze one or a type of chemical reaction in a short time and reduce the generation of by-products. Horseradish peroxidase (HRP) is a member of the large class of peroxidases. Due to its commercial availability in high purity, HRP has long been used as a novel route for enzymatic hydrogel preparation, in which, assisted by the enzyme and H_2_O_2_, the phenolic hydroxyl (Ph) groups is oxidized into polyphenols linked at the aromatic ring by C–C and C–O coupling between the Ph groups [53]. Sakai et al. coupled alginate with tyramine hydrochloride first, and then prepared alginate-Ph hydrogel with horseradish peroxidase (HRP) as catalyst [56]. The results showed that the crosslinking of Ph groups enhanced the hydrophobicity of alginate, resulted in greater adsorption of cell-adhesive protein, thus acquired cellular adhesiveness in alginate. Incorporating Ph groups into polymers for HRP-catalyzed gelation has also been proved effective in other biocompatible materials (e.g., dextran, carboxymethylcellulose, chitosan and gelatin) [57,58,59].

### 3.3. Microspheres

Gel microsphere and solid microsphere made of alginate can be obtained readily, serving as a delivery system for drugs, growth factors, cells, etc. [60,61,62,63]. Generally, alginate gel-microspheres are prepared by ionic crosslinking under aqueous conditions, which are suitable for the encapsulation of cells, growth factors and biologically active proteins [64,65,66]. Compared with gel-microspheres, alginate solid-microspheres can be prepared by emulsion-solvent evaporation technology and are mainly used to load drugs. According to the direction of reaction between divalent cation and alginate, it can be further divided into external emulsification method [67,68,69] and internal emulsification method [70,71]. In the external emulsification method, the aqueous solution of alginate and the oil phase form a water-in-oil(W/O) emulsion first, and then add the aqueous solution containing Ca^2+^ into the emulsion. Subsequently, Ca^2+^ gradually diffuses from the outside of the sodium alginate droplet into the inside, while the solidification occurs from the surface to the inside. For the internal emulsification method, CaCO_3_ powder is dispersed in the aqueous solution of sodium alginate first, and then emulsification is applied to form sodium alginate droplets containing CaCO_3_ powder. When the acidic aqueous solution is dropped into the oil phase, hydrogen ions gradually diffuse into the inside of the droplet and react with CaCO_3_ to produce Ca^2+^, and followed with Ca^2+^ gradually diffusing outwards to solidify the sodium alginate droplet from the inside to the surface.

**Table 1 marinedrugs-19-00264-t001:** Alginate-based hydrogel materials.

Materials	Cross-Linking	Methods Used	Active Ingredient	Properties	Ref
Alginate + polyacrylamide (PAAM)	Ionic and covalent cross-linking	Combining weak and strong crosslinks	/	High breaking strength (fracture energies of 9000 Jm^−2^) Expand the scope of hydrogel application	[47]
Alginate + RGD-peptide	Ionic cross-linking	The gel/sol transition of calcium alginate	RGD-peptide	Enhance cell attachment	[50]
Alginate + PNIPAm		Graft modification	/	Fast response to changes in pH and temperature	[51]
Alginate + PAM	Self-assemble	Hydrogen bonds promoted self- assembly of SA in PAM matrix	/	High mechanical strength (the tensile strength reaches 266 kPa) Self-healing property	[54]
Alginate	Enzymatically cross-linking	Horseradish peroxidase (HRP)-catalyzed oxidative crosslinking reaction	/	Controlled adhesion and proliferation of cells	[56]
Alginate + ZnO	Ionic cross-linking	By a strategy combining casting and solvent evaporation processes	/	High transparency adequate mechanical strength (Young’s modulus: From 4.14 to 4.52 MPa) Enhanced in vivo wound healing ability	[72]
Alginate + chitosan + gelatin	Covalent crosslinking	The Schiff-base reaction	Tetracycline hydrochloride	Antibacterial Enhance wound healing	[73]
Alginate + PNIPAm	Ionic cross-linking	/	Cefazolin	Monitor wound status in real-time Release the drugs on-demand	[74]
Alginate + aloe vera	Ionic cross-linking	Solvent-casting process	/	Protect the wound	[75]
Alginate + human hair keratin	Macromolecular hydrogen bonds and interfacial disulfide cross-linking	/	Doxorubicin hydrochloride	Super-high drug-loading rate Improved antitumor activity	[76]
Alginate + PAAm + poly(lactide-co-glycolide)	Ionic cross-linking	Consisting of ionically and chemically crosslinking networks	Transforming growth factor beta-3 (TGF-β3)	Higher viscoelasticity (compression modulus = 59.79 ± 1.58 kPa) Enhance cartilage repair ability	[77]
Alginate + chondroitin sulfate	Ionic cross-linking	/	Bhondroitin sulfate	Enhance bone formation in bone defects	[78]
Alginate + fullerenol	Ionic cross-linking	/	Brown adipose-derived stem cells	Induce angiogenesis Reduce oxidative stress levels	[79]
Alginate +gelatin	Enzyme-catalyzed cross-linking	HRP-catalyzed	10T1/2 and HAE cells	Shorter time for enclosed cell growth Enhance cell adhesion Maintaining on demand degradability	[57]
Alginate + PEGDA + acrylic acid	Radicals radiation	Irradiate with UV light	/	Enhanced cell migration velocity keratinocytes ingrowth	[80]
Oxidized alginate (OA) + gelatin	Covalent cross-linking	rapid cross-linking and gelation with gelatin in the presence of borax	Rat hepatocytes	Maintained cell viability	[81]
Alginate + PAM	Ionic and covalent cross-linking	Ionic cross-linking by immerse, and covalent cross-linking in a blast oven	/	Antibacterial activities Collagen deposition Granulation tissue and Angiogenesis	[82]
Alginate + chitosan + nano-HA	Polyelectrolyte cross-linking	Oppositely charged groups can be driven by electrostatic interactions	Parathyroid hormone	Enhanced osteogenic differentiation of BMSCs	[83]
Alginate derivatives + PAAM azobenzene	Self-assembly	Self-repairing based on the dynamic host–guest interaction	/	Highly stretchable and tough interpenetrating(the tensile strength = 0.015 MPa, elongation = 3465%) Self-repairing behavior under light irradiation	[84]
Alginate + pectin	Ionic cross-linking	/	Simvastatin	Promotes angiogenesis, epithelial regeneration and increased collagen deposition Speed up wound healing	[85]
Alginate + chitosan + PVA	Radiation cross-linking	Gamma-radiation	Silver NPs	Antibacterial, relieve pressure ulcers	[86]
Alginate + PVA	Physical cross-linking	Freeze-thawing method	Sodium ampicillin	High protein adsorption Antibacterial	[87]

Note: Abbreviations can be checked in the Abbreviations Part at the end of the text.

Alginate microspheres can also be prepared according to the polyelectrolyte complexation principle. For example, drip preparation, the earliest and most widely used method, is mainly to mix the drug and sodium alginate solution, drop the mixed solution into the cationic solution through syringe or microporous silicone tube, stir and solidify to produce spherical alginate microcapsules [6]. However, due to the high viscosity of sodium alginate solution, adhesion between the microcapsules occurs. With the increase of alginate concentration, the size of the prepared balloon will be more and more uniform and close to spherical, but the reaction is difficult when the concentration of sodium alginate is too high. The spray drying, the mainly used method, is to make the alginate solution form the droplets through the nozzle under high pressure [88]. When the high-pressure liquid is ejected at high speed, the droplet will be sheared and burst instantaneously, then the broken droplets are dried under high temperature by hot air and form microspheres. The principle of electrostatic granulation is that when in a high-voltage electrostatic field, the charge will be attached to the surface of the liquid. Once the flow rate and voltage exceed a specific value, the liquid will continue to grow under the action of surface electrostatic repulsion, gravity and pressure inside the liquid, and then liquid droplets are obtained [89]. The droplets drop into the Ca^2+^ containing solution at low temperature and rapidly solidify to produce microspheres.

### 3.4. Fiber

The alginate fiber prepared with seaweed as raw material has high hygroscopicity, high oxygen permeability, good biodegradability and biocompatibility, and has been widely used in medical dressings and other fields [90]. Usually, alginate fibers are prepared by wet spinning method. The soluble alginate is dissolved in water to prepare a spinning solution with a certain concentration. Squeeze the spinning solution through the spinneret into a suitable coagulation bath (usually CaCl_2_ solution), thus form insoluble alginate fibers. Zheng et al. used wet spinning to mix 2-methylimidazole zinc salt and sodium alginate and acquired alginate composite fibers with good mechanical properties, high breaking strength and antibacterial activity [91].

Alginate nanofibers can be obtained directly by electrospinning. Nanofibers have the characteristics of high specific surface area and large porosity, which can simulate extracellular matrix and is beneficial to promote the proliferation of epithelial cells and the formation of new tissues [92]. Its nano-diameter and nanofiber mesh can promote hemostasis of damaged tissues, enhance liquid absorption, promote drug delivery, cell respiration and high gas penetration, thereby prevent bacterial infections [92,93]. However, since pure sodium alginate is a polyelectrolyte, it has high conductivity in aqueous solution and needs to be blended with other polymers to prepare alginate composite fibers.

## 4. Biomedical Applications

### 4.1. Wound Healing

Currently, alginates have been widely used in wound healing due to their beneficial properties, such as biocompatibility, non-toxicity and high absorption capacity. Alginates have been prepared in various forms for wound dressings, including hydrogels, films, nanofibers and topical preparations [40,90,93]. Some of commercially available alginate dressing is in band-aid and bandages in “dressings” with alginate. Compared with traditional raw materials (such as gauze), wound dressings prepared from alginate can absorb excess wound fluid, provide a moist environment, minimize bacterial infections at the wound site, and promote wound healing (Table 2) [3]. Due to the poor mechanical properties of one component alginate material, it is usually combined with synthetic polymers to enhance the mechanical properties of the dressing. The therapeutic efficacy of composite wound dressings is affected by the component proportion of synthetic polymers, the type and degree of cross-linking, the incorporation of nanoparticles and antimicrobial agents [40].

Wound dressings provide a physical barrier between the wound and the external environment to prevent further injury or infection, and hydrogel is an ideal choice because of its non-adhesiveness, ductility and resemblance to living tissues [94]. Hydrogel can provide moisture to the wound and maintain a moist environment for cell migration. It also accelerates wound healing and reduces infection by promoting collagen synthesis, epithelial regeneration and lowering the pH of the wound [95,96]. Since chronic wounds are associated with alkaline pH, the treatment needs to restore the elevated pH to the physiological pH [5]. For the development of pH-modulating hydrogels, substances with easily ionizable groups, such as acrylic acid (AA), can be added to increase the concentration of hydrogen ions in body fluids [80]. The incorporated AA groups are responsible for pH regulation and act, together with functional groups of alginate, as interconnection between the networks. In particular, alginate hydrogels not only meet these requirements, but also activate macrophages and stimulate the late mononuclear cells to produce inter-leukin-6 (IL-6) and tumor necrosis factor-α (TNF-α) to accelerate chronic wound healing [97,98]. Wang et al. developed ZnO nanoparticles contained double-layer alginate wound dressing to prevent bacterial infection. The outer layer, i.e., a sodium alginate (SA) layer, was to prevent outside invasion, while the inner layer, which contained ZnO nanoparticles, was the absorption system. The dressing reached the maximum antibacterial rate of 68.4% without obvious cytotoxicity, and had the ability to enhance the healing in vivo [72].

Chen et al. developed a composite hydrogel dressing integrating antibacterial, biodegradable microspheres and alginate hydrogels. The alginate hydrogel used OA and carboxymethyl chitosan (CMCS) as the raw material [73]. The gelatin microspheres (GMS) loaded with tetracycline hydrochloride (TH) were prepared by emulsion cross-linking method, and the composite gel dressing was synthesized with the OA-CMCS hydrogel [73]. Compared with pure hydrogels and microspheres, the composite hydrogel dressings can continuously release the drug. In addition, the composite hydrogel dressing had strong bacteriostatic effects on *Escherichia coli* and *Staphylococcus aureus*.

Wound dressings with high flexibility, high mechanical strength and porosity have received increasing attention. Ma et al. prepared sodium alginate/graphene oxide/polyvinyl alcohol (AG/GO/PVA) nanocomposite sponge by freeze-thaw cycle and freeze-dried forming method [99]. When the GO concentration was 1 wt%, the prepared sponges had uniform and interconnected porous structure, leading to good water absorption, air permeability and mechanical properties [99]. Moreover, the presence of appropriate amount of GO could promote cell proliferation. The sponges had strong inhibitory effects on *Escherichia coli* and *Staphylococcus aureus*, and in vivo evaluations showed that the sponges enhanced wound healing.

Wound healing can be improved by giving treatment at the appropriate time, so researchers have huge interest in the so-called smart hydrogel. The hydrogel can monitor the wound environment in real time, and exhibit obvious property changes in the external environment, including small changes in temperature, pH, light, ionic strength, or enzyme environment [100,101]. Mostafalu et al. proposed an intelligent and automated flexible wound dressing by embedding thermally responsive particles into an alginate hydrogel patch and casting the patch directly onto the flexible pH sensor and heater [74]. According to the data fed back by the pH sensor, the thermal responsive release patch was activated to release antibacterial agents [74]. The prepared pH sensor patch has good flexibility, can be attached to the body and provide real-time information about the wound condition.

**Table 2 marinedrugs-19-00264-t002:** The comparison of traditional gauze and different alginate dressings.

Wound Dressing	Composition	Advantages	Disadvantages
Gauze	Purified cotton;	Cheap and easy to obtain;	Frequent changing and do not provide a moist environment for the wound [90];
Sponge	Porous PVA + alginate composite foams [102];	High porosity and surface area [102];	Need a second layer of dressing to fix [103];
	Alginate + graphene oxide +PVA [104];	Comfortable, no adhesion to the wound, and low replacement frequency;	Cannot be used for infected wounds;
Nanofiber	Alginate +PVA + ZnO [105];	Strong absorption capacity [106];	Nonadherent, require secondary dressings [40];
	Alginate+ PEO + lecithin [107];	No adhesion to the wound; Simulating ECM structure [92];	Maybe cause dehydration and dryness of the wound, and difficult to remove after using for too long [108];
Hydrogel	Alginate + chitosan [73];	Provide moisture to the wound and maintain a moist wound environment [90];	Poor mechanical properties;
	More examples, see alginate hydrogel part.	Facilitates cell migration [108].	Cause skin maceration at swollen state [109].

Note: for abbreviations, refer to Abbreviations Part at the end of the text.

Films and nanofibers made of alginate as potential materials for wound dressings have been reported by some researches. The film not only protects the wound from bacterial infections, but also improves the permeability of water vapor, oxygen and carbon dioxide, which contributes to the wound healing. However, due to their high water absorption and poor thickness, films are not useful for the wounds with excessive exudation [90]. Pereira et al. prepared novel alginate/aloe films with different proportions by solvent casting [75]. The study showed that the incorporation of aloe vera has no obvious effect on the chemical properties of the film, but improves its permeability and mechanical properties. In addition, by increasing the degradation temperature and reducing the weight loss of the film, its permeability and thermal stability are improved.

Li et al. developed a new wound dressing composed of silk fibroin, sodium alginate and strontium (Sr-loaded SF/SA blend membrane) [110]. Sr-loaded SF/SA blend film not only has good physical and chemical properties, but also exhibits water absorption, moisture permeability and good biological activity. Especially, during four days of culture in vitro, the Sr-loaded SF/SA blend films prepared by treating with 5 mg/mL Sr solution can induce a large number of basic fibroblast growth factor (bFGF) and vascular endothelial growth factor (VEGF), indicating that it can induce angiogenesis, which is very important for wound dressings [110]. In addition, some studies have shown that Sr has antibacterial activity, and its antibacterial effect is to inhibit the permeability of cell membrane, the synthesis of cell wall, the replication of bacterial chromosomes and cell metabolism by inhibiting the growth and reproduction of bacteria [111]. Therefore, the prepared new wound dressing also has antimicrobial activity.

Adding nanoparticles to the nanofibers can improve their antibacterial activity. Mokhena et al. prepared stable polyelectrolyte complex (PEC) nanofiber composites by coating chitosan/silver nanoparticles (AgNPs) onto the electrospun alginate membranes [112]. Due to their very potent and diverse antibacterial activity, AgNPs have become a popular choice of antibacterial component in polymer-based wound dressings [113]. The porous structure of PEC nanofiber composites accelerates absorbing water which in turn helps releasing the AgNPs into the medium [112]. The AgNP aggregates are well dispersed on the surface of electrospun nanofibers, thus the obtained nanofiber composite material is highly antibacterial against gram-negative bacteria and gram-positive bacteria. However, it cannot be released for a long time. Hajiali et al. used sodium alginate and lavender essential oil to produce bioactive nanofiber dressings by electrospinning [114]. The results showed that the addition of lavender oil did not affect the morphology of the nanofibers, and the nanofibers prepared were highly hydrophilic. The in vivo studies have shown that sodium alginate and lavender essential oil nanofiber dressings not only have antibacterial activity against *Staphylococcus aureus*, but also effectively inhibit the production of pro-inflammatory cytokines in vivo and in vitro.

### 4.2. Drug Delivery

Alginate plays an important role in drug delivery. At present, the oral dosage form uses alginate the most frequently in drug application, and alginate as a carrier for local drug delivery has been paid more and more attention [2]. Encapsulating certain active substances (nucleic acids, cells, enzymes, proteins, drugs, etc.) in the alginate substrate can protect the drugs, prevent the premature inactivity of the drugs, delay the release of the drugs, and enable the drugs to reach the target site at a fixed point and at a fixed time to complete the targeted therapy. A schematic presentation of alginate-based material preparation for cancer therapy is shown in Figure 4. Shaedi et al. designed an oral gut specific alginate nano-system for vitexin [115]. In the study, stearic acid was used to make the matrix hydrophobic, which promotes the early release of vitexin, and the nanoparticles are compacted with polyethylene glycol (PEG3000, 10,000 and 20,000). The results showed that compacting the nanoparticles with PEG significantly reduced the release of vitexin in the gastric region, while the release of vitexin in the intestinal tract increased by the nanoparticles loaded with stearic acid [115]. The use of PEG-10,000 during the compaction process will lead to PEG-nanoparticle interactions, thereby inhibiting the initial release of vitexin. Then the dissolution of PEG in the subsequent intestinal phase causes the dispersed stearic acid to induce particle rupture and vitexin release. PEG compressed nanoparticles showed specific release of vitexin by oral intestines, decreased positive blood glucose in the body and increased intestinal vitexin content.

Sun et al. prepared a novel double-stimulation-responsive nanogel using human hair keratin and alginate as raw materials through a simple cross-linking method [76]. Keratin provides the cross-linking structure and biological reactivity of nanogels, while alginate improves the properties of nanogels, such as particle size, stability and drug loading capacity [76]. The prepared keratin-alginate nanogel (KSA-NGS) has a high drug loading rate (52.9%) and can effectively load and deliver doxorubicin hydrochloride (DOX) to cancer cells, which can effectively inhibit the occurrence of tumor. In in vivo experiments, it has shown that KSA-NGS gel loaded with DOX aggregates more easily in tumors, stays longer, has better antitumor activity and fewer side effects compared to the gel free of drugs.

Alginate can incorporate protein into alginate-based materials under relatively mild conditions, minimizing its denaturation and making it unaffected by the acid environment of the stomach. Due to the inherent porosity and hydrophilicity of the gel, the release of protein from the alginate gel is very fast. However, if the encapsulated protein is positively charged, then the protein can interact with the negatively charged sodium alginate, inhibiting its diffusion and release in the polymer network, and achieving sustained and local release. This release can also be controlled by changing the degradation rate of alginate. Mata et al. applied a combination of polylactic acid (PLGA) and sodium alginate to vaccine delivery [104]. As the main substrate, PLGA could avoid the instability of sodium alginate hydrogel in the physiological environment, while sodium alginate can improve the encapsulation rate of PLGA particles and significantly reduce the initial burst release [104]. Studies have shown that the prepared PLGA/Alg carrier microspheres have higher encapsulation efficiency and immune-mediated ability than a single carrier.

Zhang et al. used functionalized alginate (ALG) nanoparticles to deliver targeted antigen to dendritic cells for cancer immunotherapy [118]. Mannose (MAN) modified alginate (MAN-ALG) was used for DC targeting and the MAN-ALG/ALG = OVA nanoparticles (MAN-ALG/ALG = OVA NPs) were prepared via crosslinking MAN-ALG and ALG = OVA by CaCl_2_. The results showed that MAN-ALG/ALG = OVA NPs facilitated antigen uptake of BMDCs and cytosolic release of the antigen. In vivo studies have shown that nanoparticles can be effectively transported from the injection site to the draining lymph nodes. In addition, Man-Alg/Alg = OVA NPS can enhance the cross expression of OVA in B3Z T-cell hybridoma, and subcutaneous administration of MAN-ALG/ALG = OVA NPS in mice also induced cytotoxic T lymphocyte (CTL) response and inhibited the growth of EG7 tumors.

Zhang et al. modified alginate (AlG) nanoparticles (DOX/ GA-AlG NPS) with glycyrrhizinic acid (GA) loaded with doxorubicin (DOX) [119]. In the DOX/ GA-Alg NPS group, the growth inhibition rate (IR) of orthotopic liver tumors was 76.6%, with no mouse death, compared to approximately 52.6% and 33% mortality in the control group. This indicates that DOX/ GA-AlG NPS can effectively inhibit the growth of liver tumors in situ. Most importantly, DOX/GA-ALG NPs had no effect on the heart and liver cells around the tumor and reduced side effects significantly.

### 4.3. Tissue Repair and Regeneration

Tissue engineering can combine cells and biomaterials to reconstruct the structure, shape and function of damaged tissues and organs, thus replace damaged tissues. Seed cells, growth factors and scaffolds are three elements of tissue engineering technology [3]. One of the earliest applications of sodium alginate in tissue engineering is to encapsulate pancreatic islet grafts in sodium alginate hydrogel for the treatment of diabetes. The advantage of sodium alginate as a scaffold material is that it can fully adapt to tissue defects because of the excellent flexibility [7], and simultaneously load bioactive molecules as well [120]. The unique three-dimensional gel structure provides comfortable stereo space for the growth of the seed cells, and the shape is regular and the surface is smooth, which can avoid secondary damage to the injured site when the implant of irregular shape is implanted. It has been used in the cartilage [120], hard bone [121], nerve tissue repair [122,123] and other aspects [124].

Saygili et al. prepared a functionalized polyacrylamide (PAAM)-ALG double network hydrogel [77]. Subsequently, the PLGA NPs loaded with functional transformation factors were encapsulated in this hydrogel, similar to articular tissue, and maintained their mechanical stability for over 3 months at different temperatures (+4, 25, 40 °C) and humidity conditions (60% and 75%). In vitro experiments showed that, compared with PAAM-ALG hydrogels, the functional hydrogel exhibited better cell viability and significantly promoted the regeneration of rat cartilage. Witte et al. designed an alginate-fibronectin microfluidic carrier (known as a cartilage bag) equipped with solid presentation of growth factors capable of preserving the human articular chondrocyte phenotype and promoting chondrogenic differentiation of skeletal stem cells [125]. The results demonstrated the biocompatibility, cell viability, proliferation and tissue-specific differentiation of chondrocyte markers. It illustrated the potential applications for a TGF-β1 alginate-fibronectin chondro-bags platform as a workable 3D bioprinting and culture system for cartilage tissue regeneration with therapeutic applications therein.

Alginate hydrogel has many advantages, such as its inherent biocompatibility, high water content and molecular structure similar to natural extracellular matrix, so it has a great application potential in bone regeneration and bone defect repair. Ma et al. prepared strontium alginate (Sr) hydrogel containing chondroitin sulfate (CS) for enhanced bone defect repair [78]. Studies have shown that strontium (Sr), a trace element in bone, has a positive effect on bone regeneration, can enhance the proliferation and differentiation of osteoblasts, and reduce the activity of osteoclasts [126]. As an important multifunctional sulfide GAG, CS can participate in the process of bone formation and mineralization. The results of in vitro experiments showed that the SR-CS/alginate gel with higher CS ratio was beneficial to the proliferation of osteoblasts, and the Sr-CS/alginate gel had a positive regulation effect on osteogenic factors.

Bone repair is very slow and it is difficult to regain full function. This is mainly due to the existence of a lot of reactive oxygen species (ROS)/free radicals at the fracture site. Bone injury and surgical trauma will produce oxidative stress in the damaged tissue, thus increasing the generation of free radicals [127]. The presence of free radicals during healing further delays the healing process, and high levels of free radicals can damage cells through protein and lipid oxidation. In addition, it can alter DNA and mitochondrial integrity, or trigger apoptosis of bone cells [128,129]. Purohit et al. mixed nano-cerium oxide (Nanoceria, NC) into gelatin-alginate (GA) scaffolds and obtained nano-composite scaffolds (GA-NCs) by freeze-drying. NC has good free radical scavenging ability [130], the results showed that the addition of NC increased the mechanical properties and biomineralization of the scaffold, and reduced the expansion and weight loss of the scaffold. The synergistic fusion of nanoparticles and GA scaffolds enhanced the adhesion, proliferation and activity of cells, making GA-NCS scaffolds have the potential to assist the differentiation of mesenchymal stem cells (MSCs) into osteoblasts, and have a certain ability of free radical scavenging. In order to solve the problem of low retention rate and survival rate of stem cells after transplantation ascribing to the existence of reactive oxygen species (ROS) microenvironment. Tong et al. introduced fullerenol nanoparticles into alginate hydrogels to create an injectable cell delivery vehicle with antioxidant activity [79]. The results showed that the prepared fullerenol/alginate hydrogels had good injection strength and mechanical strength, and could effectively scavenge superoxide anions and hydroxyl radicals. Fullerenol/alginate saline gel had no cytotoxic effect on the biological behaviors of brown adipose-derived stem cells (BADSCs). It could effectively reduce the ROS level in the myocardial infarction (MI) region, improve the retention rate and survival rate of implanted BADSCs, induce angiogenesis, and thus promote the recovery of cardiac function. The mechanism of action was to inhibit oxidative stress damage of BADSCs via activating ERK and p38 pathways while inhibiting JNK pathways, and improve its survival ability in ROS microenvironment.

### 4.4. 3D Bioprinting

3D bioprinting, also known as biofabrication, is a new additive manufacturing technology for fabrication of structures resembled in architecture to native biological tissue. Compared with non-biological printing, 3D bioprinting enables 3D printing of biocompatible materials, cells and supporting components into complex 3D functional living tissues. Under the banner of robotic rapid prototyping, 3D bioprinting has emerged as a potential tool in regenerative medicine since last decade, and is currently being applied to fabricate 3D functional constructs with biological and mechanical properties suitable for clinical restoration of tissue and organ function.

Most of the 3D bioprinting researches have been conducted for the applications in bone and cartilage regeneration. Antich et al. prepared hyaluronic acid (HA) and alginate (ALG) hydrogel based bioink [108]. The mixtures of HA and ALG provide the mechanical properties suitable for cell laden. The printed bioink/PLA composite scaffold can support cartilage extracellular matrix deposition and gene expression in vitro. With 3D bioprinting technique, Wu et al. constructed a gelatin/sodium alginate hydrogel scaffold for neural repair with rat Schwann cells contained [131]. After cultured in the hydrogel for seven days, it was found that the cells had a high survival rate and adhered to the surface of the scaffold firmly. Compared with the 2D culture samples, 3D bioprinted samples showed higher mRNA levels of NGF, brain-derived neurotrophic factor (BDNF), glial neurotrophic factor (GDNF) and platelet-derived growth factor (PDGF) on the fourth day. These results suggest that the composite scaffold can maintain the activity of Schwann cells and promote the expression of cell adhesion and related factors.

The cells used in 3D bioprinting should be kept to the physiological state and maintained with physiological function [132], and normally, need to be expanded in vitro till enough for printing. Over the years, bioink has been the focus of the research within 3D bioprinting field. Defined as a formulation suitable for processing by an automated biofabrication technology, bioink may also contain biologically active components and biomaterials [132]. Compared with directly loaded cell hydrogel, the application of scaffold free cell gel can obtain higher cell density. Without waiting for cell proliferation, it can stimulate cells to synthesize ECM, greatly improve the efficiency and accuracy of biological 3D printing process [133]. Fedorovich et al. successfully demonstrated the possibility of manufacturing viable centimeter-scaled structured tissues by the 3D fiber deposition technique [134]. In this study, the fluorescently labeled human chondrocytes and osteogenic progenitors were encapsulated and printed in alginate hydrogel yielding scaffolds of 1 × 2 cm with different parts for both cell types. Cell viability remained high throughout the printing process, and cells remained in their compartment of the printed scaffold for the whole culture period. Moreover, distinctive tissue formation was observed, both in vitro after three weeks and in vivo (six weeks subcutaneously in immunodeficient mice), at different locations within one construct. In addition, some studies have shown that incorporating the tissue-specific ECM into bioink can promote the specific differentiation of the cells. Choi et al. prepared the decellularized skeletal muscle extracellular matrix (mdECM)-based bioink [135]. These mdECM-based bioinks can be extruded at 4 °C and then fabricated and solidified by raising the temperature to 37 °C. The results showed that the cells encapsulated in these bioinks remained active after printing, the mdECM bioink provideed the 3D cell-printed muscle constructs with a myogenic environment that supports high viability and contractility as well as myotube formation, differentiation, and maturation.

## 5. Conclusions and Prospect

Alginate, which is rich in sources, and has excellent biocompatibility, biodegradability, non-toxicity and safety, has been widely used in drug delivery carriers, medical wound dressings, skeleton materials and delivery of bioactive substances in tissue engineering, etc. In this paper, the research status of alginate medical materials in different fields was reviewed in detail. Compared with existing medical materials, alginate wound dressings have higher water absorption and porosity, continuous drug release ability and non-immunogenicity, which can not only maintain a certain degree of moisture for the wound microenvironment, but also accelerate the speed of epithelial regeneration, granulation tissue formation and wound healing. However, most of the existing alginate medical dressings do not have intelligent characteristics and cannot monitor wound conditions in real time, which hinders their application. Therefore, the development of smart alginate medical materials with new structures will become the focus of research in the future.

While alginate is widely used in tissue engineering, there are still many problems to be solved: How to improve the mechanical properties and biocompatibility of hydrogels; how to promote the adhesion, proliferation and differentiation of osteoblasts, and promote the angiogenesis and the formation of bone tissue; how to control the porosity of the hydrogel to ensure the transportation of nutrients and metabolites, as well as cell proliferation and transfer; and how to provide enough oxygen for the microencapsulated cells in the scaffold and avoid cell death. The alginate drug carrier has targeted therapeutic effect, and the drug is released slowly and uniformly in the body, which improves the therapeutic effect of the drug. However, how to realize dynamic and intelligent drug release, and how to carry out continuous and sequential release according to external signals, remains to be solved. Alginate fiber materials have great advantages in the field of biology, but there are some problems, such as poor mechanical properties, weak binding force between fibers, uncontrollable hygroscopicity and single function. Therefore, for the development of alginate fiber in the future, it is necessary to optimize the production process, improve the mechanical properties, optimize the design of functional alginate fiber, and develop multi-functional alginate fiber products.

## Figures and Tables

**Figure 1 marinedrugs-19-00264-f001:**
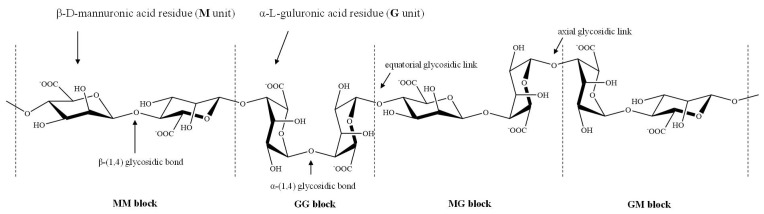
Stylized conformation structures of alginate units, blocks and their linkages M unit: β-d-mannuronic acid residues; G unit: α-l-guluronic acid residues; MM block: homopolymeric blocks of M units; GG block: homopolymeric blocks of G units; and MG or GM block: heteropolymeirc blocks of M and G or G and M units.

**Figure 2 marinedrugs-19-00264-f002:**
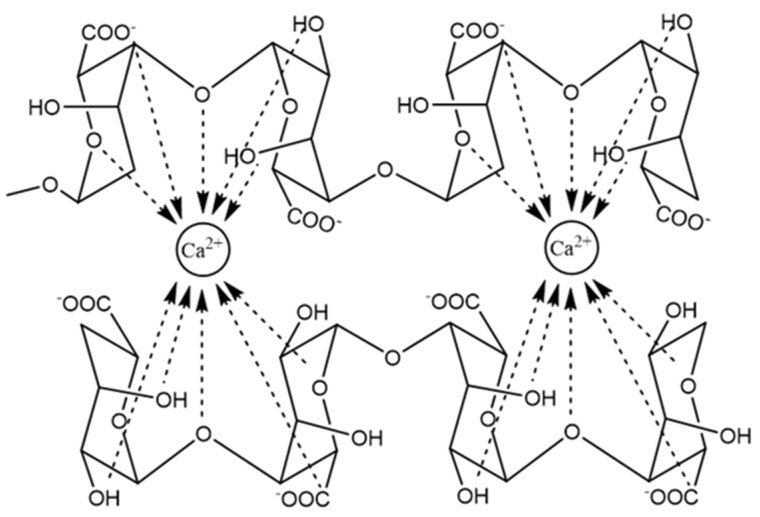
The junction zone in the egg-box model of calcium alginate gel.

**Figure 3 marinedrugs-19-00264-f003:**
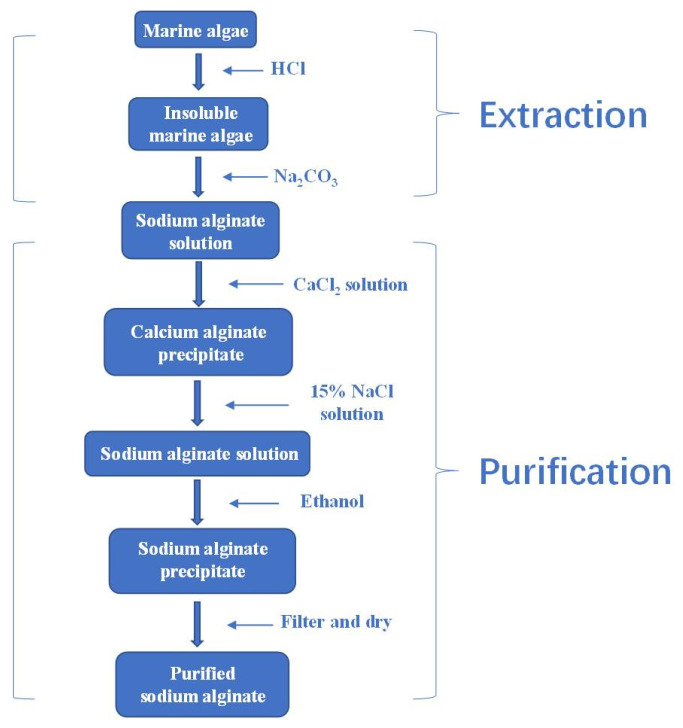
The industrial process of sodium alginate extraction via calcium precipitation.

**Figure 4 marinedrugs-19-00264-f004:**
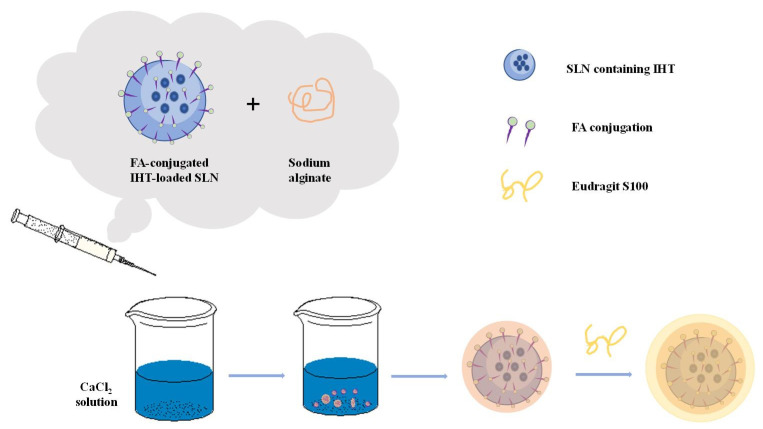
An example of alginate-based material for cancer therapy. Schematic preparation of folic acid (FA)-grafted solid lipid nanoparticles (SLNs) bearing irinotecan hydrochloride trihydrate (IHT) are encapsulated in alginate matrix coated with Eudragit S100 [116,117].

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
