# Peer review of "Preparation of Alginate-Based Biomaterials and Their Applications in Biomedicine"

_marinedrugs, 2021, doi:10.3390/md19050264_

Round 1
Reviewer 1 Report
The overall subject of alginate use for biomedical purposes is interesting.
However, the current review has some major flaws and lack of some critical key details that need to be addressed carefully and thoroughly before the paper can be endorsed for publication
1. The English is poor. The past tense has been used a lot - but has been used in a wrong way (e.g. "This review introduced the production of alginates-.."?). You need to have a professional scientific editor to work through
2. The review sidesteps too many of the key intriguing points regarding the chemistry, hydrocolloid mechanism for good gelation, and avoids a proper educational discussion of the gelation mechanisms. Hydrocolloid alginate gelation has recently been discussed in a review in Trends in Food Science and Technology.
2a. Especially: Fig 1 is just too light and if the gelation is not properly discussed then the review has no scientific value. You need to add more generic data on alginate to make it better. Novelty is already a concern as alginate use in biomedical applications is not novel.
2b. Specifically: Please address / discuss in more depth that it is known that some Sargassum alginates have favorable, low M/G ratios (below 1) and
high contents of GG-blocks, but nevertheless Sargassum alginates often show poor gelling properties (check e.g. Mchugh, D. J. (2003). A guide to the seaweed industry. FAO Fisheries Technical Paper No. 441.) - This is a huge issue in alginate physics chemistry. On the other hand alginate from Padina sp. with a high M/G ratio have been shown to produce good gels (https://doi.org/10.1016/j.foodhyd.2017.05.016) so it is important to discuss the chemistry and not only the M/G ratio.
3. The review has a few interesting postulates for the alginate extraction that are wrong/not followed up or substantiated: e.g. p. 2 line 55-56: "Nowadays enzymes are often used to replace the acid because it dan reduce the caused degradation"...? Would be good to have a reference here - enzyme assisted alginate extraction is not generally known?
3a. FIGURE 1: "Digestion" ? Digestion is not the right term. Line 62: You would not add HCl to have the alginate precipitate!? Figure 1 Decalcification with NaCl chromatography? That needs a reference
4. Discuss the difference in chemistry between bacterial alginate and alginate from macroalgae. Rephrase line 65-70,
5. Table 1 and text on p. 4 about "gelation" : You need to add more details about the desirable gel-strength (Pa-values) and measureable viscoelastic properties. If not it is not convincing why alginate properties and utility for biomedical purposes should be particularly attractive compared to other hydrocolloids.
6. Table 1: "Horseradish peroxidase catalyzed cross-linnking of alginate"? Discuss the mechanisms, there are two such entries in Table 1, and that needs explananation - What happens to the enzyme if it is used in biomedical applications?
7. Table 1: Covalent cross-linking with borate? Are you sure? It may rather be an ionic binding? (entry line 7 from bottom in the table). In general the entries in the Table are just listed, but not properly discussed or detailed, so the review becomes very superficial when things are just listed, the text is light, and there is no proper discussion of phenomena, mechanisms, and gel strength
8. It is not for "biomedicine", but for biomedical purposes. Alginate is not the medicine itself.
9. Figure 3. It is not scientifically valid to just compare with some words. What kind of chemistry, alginate origin, extraction and modifications are required for specific performance in the different applications?
10. line 229-230 By what mechanism would alginate be bacteriostatic Agains both Gram+ and Gram- organisms?
11. Please include a Figure about drug delivery for clarity - The text is not easy to read - and what is the alginate compatibility to cancer cells?
Author Response
- The English is poor. The past tense has been used a lot - but has been used in a wrong way (e.g. "This review introduced the production of alginates-.."?). You need to have a professional scientific editor to work through
Response: The English writing in the revised paper has been checked thoroughly and carefully. We will keep improving our English.
- The review sidesteps too many of the key intriguing points regarding the chemistry, hydrocolloid mechanism for good gelation, and avoids a proper educational discussion of the gelation mechanisms. Hydrocolloid alginate gelation has recently been discussed in a review in Trends in Food Science and Technology
Response: A whole part on alginates' chemical compositions, structures and properties has been added in the revised paper. In Part 2, the chemistry, gelling mechanism and recent advances within this field have been discussed. By the way, thanks very much for the recommendation.
- Especially: Fig 1 is just too light and if the gelation is not properly discussed then the review has no scientific value. You need to add more generic data on alginate to make it better. Novelty is already a concern as alginate use in biomedical applications is not novel.
Response: As the whole part on the composition , structure and properties of alginates has been added in the revision, Fig 1 has been deleted and replaced by other diagram suitable to the context. The generic knowledge has been supplemented and the gelation has been discussed thoroughly within this part. The recent progress in the gelation mechanism research is also reviewed.
- Specifically: Please address / discuss in more depth that it is known that some Sargassum alginates have favorable, low M/G ratios (below 1) and high contents of GG-blocks, but nevertheless Sargassum alginates often show poor gelling properties (check e.g. Mchugh, D. J. (2003). A guide to the seaweed industry. FAO Fisheries Technical Paper No. 441.) - This is a huge issue in alginate physics chemistry. On the other hand alginate from Padina sp. with a high M/G ratio have been shown to produce good gels (https://doi.org/10.1016/j.foodhyd.2017.05.016) so it is important to discuss the chemistry and not only the M/G ratio.
Response: Thanks very much for your guide. The related discussion has been added in Part 2. We hope it has been elaborated clearly in the present paper.
- The review has a few interesting postulates for the alginate extraction that are wrong/not followed up or substantiated: e.g. p. 2 line 55-56: "Nowadays enzymes are often used to replace the acid because it dan reduce the caused degradation"...? Would be good to have a reference here - enzyme assisted alginate extraction is not generally known?
Response: We especially add a paragraph on enzyme assisted extraction in Part 2. The reference is also provided.
- 3a. FIGURE 1: "Digestion" ? Digestion is not the right term. Line 62: You would not add HCl to have the alginate precipitate!? Figure 1 Decalcification with NaCl chromatography? That needs a reference
Response: Fig 2 is replaced by a flow diagram about the industrial sodium alginates production process. In the context, we mean the alginc acid can be precipitated by adding HCl in the sodium alginate solution. We rewrite the sentence so as to avoid the confusion.
- Discuss the difference in chemistry between bacterial alginate and alginate from macroalgae. Rephrase line 65-70,
Response: The whole paragraph on bacterial alginates has been rewritten including the difference in chemistry between bacterial alginate and alga alginate.
- Table 1 and text on p. 4 about "gelation" : You need to add more details about the desirable gel-strength (Pa-values) and measureable viscoelastic properties. If not it is not convincing why alginate properties and utility for biomedical purposes should be particularly attractive compared to other hydrocolloids.
Response: The corresponding data have been supplemented in Table 1
- Table 1: "Horseradish peroxidase catalyzed cross-linking of alginate"? Discuss the mechanisms, there are two such entries in Table 1, and that needs explanation - What happens to the enzyme if it is used in biomedical applications?
Response: The mechanism of HRPcatalyzed cross-linking has been supplemented in the context. HRP assists the oxidization of phenolic hydroxyl into polyphenols, thus incorporating Ph groups into polymers.
- 7. Table 1: Covalent cross-linking with borate? Are you sure? It may rather be an ionic binding? (entry line 7 from bottom in the table). In general the entries in the Table are just listed, but not properly discussed or detailed, so the review becomes very superficial when things are just listed, the text is light, and there is no proper discussion of phenomena, mechanisms, and gel strength
Response: According to the reference, sodium alginate was periodate oxidized, thus incorporating the dialdehyde group on the polymer chain. Only small amount of Borax was in the system to assist rapid and injectable gelling. Borax can react with oxidized alginate by complexing with hydroxyl groups of polysaccharides. With the slightly alkaline pH of the medium, oxidized alginate crosslink with gelatin via Schiff’s base formation between the e-amino groups of lysine or hydroxylysine side groups of gelatin and the available aldehyde on oxidized alginate.
The data of gel strength have been supplemented in the Table1.
- 8. It is not for "biomedicine", but for biomedical purposes. Alginate is not the medicine itself.
Response: The sentence has been rephrased.
- 9. Figure 3. It is not scientifically valid to just compare with some words. What kind of chemistry, alginate origin, extraction and modifications are required for specific performance in the different applications?
Response: Fig 3 has been deleted and replaced with Table 2. As the modifications for different applications are actually based on essential chemistry, we didn't add the content in detail. And presently most of the alginates applied in the research for biomedical applications are commercially algal alginates, so we didn't note the origin.
- line 229-230 By what mechanism would alginate be bacteriostatic Agains both Gram+ and Gram- organisms?
Response: The mechanism is shortly explained in the revised context. Ag Nano particles exhibit unselective binding to the proteins of bacteria and make the proteins denatured, thus alginates containing AgNP can be bacteriostatic against both Gram+ and Gram- organisms..
- Please include a Figure about drug delivery for clarity - The text is not easy to read - and what is the alginate compatibility to cancer cells?
Response: Fig 4illustrating the Alginate-based material for cancer therapy is added for clarity. The compatibility of alginate to cancer cell needs to be researched. Generally speaking, purified alginates don't show toxicity to any cell.
Reviewer 2 Report
The current manuscript mentioned the characteristics of advantage of wound healing formulation. The alginate-incorporated hydrogel, sponge, film, and fiber were introduced in the current manuscript. The alginate-based oral drug delivery system, nanoparticulate systems, tissue engineering containing bone generation was introduced. I have enjoyed the manuscript. I left several comments.
-If it is possible, can you add the part of abbreviation for the difficulty of tracing?
-Can you increase the contents of 3D bioprinting and the optimal properties of 3D printer ink (e.g. viscosity), if it is possible? Because the attention about 3D bioprinting is increasing.
-L.213 lowering the pH of the wound.
More explanation about the reason is necessary
-As various polymers and compounds were mixed with alginate which has negative charge, can you explain the interaction with polymer or drug, if it is possible. For example, the reviewer thinks that the interaction of alginate with compound of polymer with cationic charge may change the viscosity and drug release remarkable. Can you introduce the information about it?
-Are there any studies about the immune response about alginate?
Author Response
-If it is possible, can you add the part of abbreviation for the difficulty of tracing?
Response: Yes, the part of abbreviation has been added at the end of the revised paper.
-Can you increase the contents of 3D bioprinting and the optimal properties of 3D printer ink (e.g. viscosity), if it is possible? Because the attention about 3D bioprinting is increasing.
Response: The content of 3D bioprinting has been supplemented. As the requirements on the bioink for different applications are different, even for different machine , the parameters can vary greatly, it's hard to standardize or normalize at present.
-L.213 lowering the pH of the wound. More explanation about the reason is necessary
Response: The explanation about lowering the pH of the wound has been provided in the context.
-As various polymers and compounds were mixed with alginate which has negative charge, can you explain the interaction with polymer or drug, if it is possible. For example, the reviewer thinks that the interaction of alginate with compound of polymer with cationic charge may change the viscosity and drug release remarkable. Can you introduce the information about it?
Response: Normally, the drugs will be loaded with more stable polymer for long term release before encapsuled by alginates. When mixed with alginate,CaCl2 solution works as the base, the whole system is in balance, thus the effect of charge on the alginate polymer will not change the drug release. Fig 4 is supplemented which can show the process more clearly.
-Are there any studies about the immune response about alginate?
Response: Yes, the studies about immune response to alginates have been reviewed and added in Part 2.
Round 2
Reviewer 1 Report
The revised version provides adequate responses to the issues raised in the first review-round. A few items:
- The English would benefit from professional English proof-reading (especially in the first parts of the paper), the revised text has some peculiar linguistic mistakes that disturbs the reading.
- Line 288: The Strontium vs. Calcium. Since Strontium-90 is known to be radio-active, you should add that "as long as Strontium only has stable isotopes (naturally occurring), the health risk is minimal.
- Line 325-333: The use of horse-radish peroxidase for Alginate gelation is highly disputable - it is uncertain and possibly wrongly explained in ref 53 how the gelation occurs. The statement line 332-333 is true, but the majority of HRP-catalyzed gelations is on PECTIN via ferulate-oxidation that directly cross-links, so you must take this section out as it has no relation to alginate.
- Section 4.1. Please note, alginate is already used commercially in band-aid and bandages in "dressings" with alginate.
Author Response
1. The English would benefit from professional English proof-reading (especially in the first parts of the paper), the revised text has some peculiar linguistic mistakes that disturbs the reading.
Response: Section 1 has been rephrased. The English in the revised paper has been checked thoroughly, several minor revisions have been made. Hope the present paper is more readable.
2. Line 288: The Strontium vs. Calcium. Since Strontium-90 is known to be radio-active, you should add that "as long as Strontium only has stable isotopes (naturally occurring), the health risk is minimal.
Response: The illustration has been added in Line 291.
3. Line 325-333: The use of horse-radish peroxidase for Alginate gelation is highly disputable - it is uncertain and possibly wrongly explained in ref 53 how the gelation occurs. The statement line 332-333 is true, but the majority of HRP-catalyzed gelations is on PECTIN via ferulate-oxidation that directly cross-links, so you must take this section out as it has no relation to alginate.
Response: We are sorry for not having supplemented enough details about alginate hydrogel preparation via HRP enzyme crosslinking in Ref 53. The first step is the coupling of alginate with tyramine hydrochloride under the catalysis of NHS and EDC, thereby introducing the Ph group on alginate polymer. Then mix the obtained Alg-Ph solution with H2O2 solution and HRP solution, incubate the mixture at 37℃ and the gel forms. During the crosslinking, HRP catalyzes the oxidation of phenolic hydroxyl (Ph) groups, resulting in polyphenols linked at the aromatic ring by C-C and C-O coupling between the Ph groups
4. Section 4.1. Please note, alginate is already used commercially in band-aid and bandages in "dressings" with alginate.
Response: The band-aid applications of alginates has been supplemented in Section 4.1.
